# Prediction of Surgical Outcome in Advanced Ovarian Cancer by Imaging and Laparoscopy: A Narrative Review

**DOI:** 10.3390/cancers15061904

**Published:** 2023-03-22

**Authors:** Patrícia Pinto, Andrea Burgetova, David Cibula, Ingfrid S. Haldorsen, Tereza Indrielle-Kelly, Daniela Fischerova

**Affiliations:** 1Department of Gynecology, Portuguese Institute of Oncology Francisco Gentil, 1099-023 Lisbon, Portugal; aplpinto@gmail.com; 2First Faculty of Medicine, Charles University and General University Hospital in Prague, 121 08 Prague, Czech Republic; 3Department of Radiology, First Faculty of Medicine, Charles University and General University Hospital in Prague, 121 08 Prague, Czech Republic; andrea.burgetova@vfn.cz; 4Gynecologic Oncology Center, Department of Obstetrics and Gynecology, First Faculty of Medicine, Charles University and General University Hospital in Prague, 121 08 Prague, Czech Republic; david.cibula@vfn.cz; 5Mohn Medical Imaging and Visualization Centre, Department of Radiology, Haukeland University Hospital, 5009 Bergen, Norway; ingfrid.haldorsen@uib.no; 6Section of Radiology, Department of Clinical Medicine, University of Bergen, 5021 Bergen, Norway; 7Department of Obstetrics and Gynaecology, Burton and Derby Hospitals NHS Trust, Derby DE13 0RB, UK; tereza.indrielle-kelly@nhs.net

**Keywords:** ovarian neoplasms, ultrasonography, multidetector computed tomography, diffusion magnetic resonance imaging, positron emission tomography computed tomography, neoplasm staging, laparoscopy

## Abstract

**Simple Summary:**

Maximal-effort debulking surgery is the recommended approach for advanced-stage ovarian cancer. The role of imaging is to provide a preoperative systematic and structured report of tumour dissemination with special emphasis on key sites that preclude optimal resectability in ovarian cancer surgery. Imaging methods cannot reliably detect small volume carcinomatosis but yield high diagnostic performance for detecting bulky disease at critical sites for cytoreduction and can thus be reliably used to avoid unnecessary explorations. Although diagnostic laparoscopy may directly visualize intraperitoneal involvement, it has inherent limitations when investigating tumours behind the gastrosplenic ligament, in the lesser sac, mesenteric root or when exploring the retroperitoneum. The major benefit of laparoscopy appears as an ultimate triage step in situations where the imaging diagnosis is uncertain regarding resectability and the presence of diffuse small-volume carcinomatosis.

**Abstract:**

Maximal-effort upfront or interval debulking surgery is the recommended approach for advanced-stage ovarian cancer. The role of diagnostic imaging is to provide a systematic and structured report on tumour dissemination with emphasis on key sites for resectability. Imaging methods, such as pelvic and abdominal ultrasound, contrast-enhanced computed tomography, whole-body diffusion-weighted magnetic resonance imaging and positron emission tomography, yield high diagnostic performance for diagnosing bulky disease, but they are less accurate for depicting small-volume carcinomatosis, which may lead to unnecessary explorative laparotomies. Diagnostic laparoscopy, on the other hand, may directly visualize intraperitoneal involvement but has limitations in detecting tumours beyond the gastrosplenic ligament, in the lesser sac, mesenteric root or in the retroperitoneum. Laparoscopy has its place in combination with imaging in cases where ima-ging results regarding resectability are unclear. Different imaging models predicting tumour resectability have been developed as an adjunctional objective tool. Incorporating results from tumour quantitative analyses (e.g., radiomics), preoperative biopsies and biomarkers into predictive models may allow for more precise selection of patients eligible for extensive surgery. This review will discuss the ability of imaging and laparoscopy to predict non-resectable disease in patients with advanced ovarian cancer.

## 1. Introduction

Ovarian cancer is the most common cause of death from gynaecological cancer [1]. It is often diagnosed at an advanced stage with extensive peritoneal and/or distant metastases, which reduce the survival rate to 20–40% in stage IIIC and 10% in stage IV according to the International Federation of Gynecology and Obstetrics (FIGO) classification [2]. Population-based screening is ineffective, and new approaches to early diagnosis and prevention using molecular genomics are still in development [3,4,5,6]. Patient prognosis is reportedly improved if the treatment is provided at an accredited gynaecologic oncology centre (Figure 1) [7,8,9,10]. To expedite the referral of ovarian cancer patients to centralized multidisciplinary care, the European Society of Gynaecological Oncology (ESGO), the International Society of Ultrasound in Obstetrics and Gynecology (ISUOG), the International Ovarian Tumour Analysis (IOTA) group and the European Society for Gynaecolo-gical Endoscopy (ESGE) have issued an evidence-based consensus statement on the preoperative diagnosis of ovarian cancer to help differentiate between benign and malignant ovarian tumours [11]. It has been shown that ultrasound detection of ovarian cancer using the IOTA models is highly accurate in discriminating between benign disease and stage I-II primary ovarian cancer [12]. Determining the presence of advanced disease when presented with ascites and peritoneal lesions is usually not a diagnostic problem.

According to the current evidence, maximal-effort upfront debulking surgery in gynaecologic oncology centres, followed by platinum-based chemotherapy and maintenance treatment as indicated, yields the best results with acceptable morbidity [13,14,15,16,17,18,19,20]. The outcomes of surgical treatment of ovarian cancer are divided into three prognostic groups based on the residual disease: (1) complete cytoreduction without macroscopic disease; (2) optimal cytoreduction with residual macroscopic disease up to 1 cm; and (3) suboptimal cytoreduction with macroscopic disease greater than 1 cm [21]. Residual disease, along with the type of systemic therapy, are the most important prognostic factors that can be influenced by the treating physician [22]. The 3-year overall survival for patients according to cytoreduction status was 72.4% (complete resection), 65.8% (residual tumour ≤ 1 cm) and 45.2% (residual tumour > 1cm), respectively, in a combined analysis of three multicentre phase III trials (AGO-OVAR 3, 5, and 7) [14,23]. The goal of upfront cytoreduction should always be complete resection, especially in mucinous and clear-cell carcinoma where no benefit has been shown for residual disease ≤1 cm [14,24,25,26]. However, small residual disease (i.e., ≤1 cm) may be acceptable in low-grade serous carcinoma, and possibly in tumours highly responsive to systemic treatment, such as high-grade serous tubo-ovarian carcinoma [26,27,28,29]. Grabowski et al. demonstrated a greater effect on survival between partial debulking to small residual disease up to 1 cm or large residual disease in low-grade serous carcinoma compared with high-grade serous carcinoma (hazard ratio (HR), 0.514 [95% CI, 0.258 to 1.022] vs. HR, 0.809 [95% CI, 0.537 to 1.220]) [28]. In contrast to upfront debulking surgery, patients who underwent neoadjuvant chemotherapy appeared to benefit from subsequent interval debulking surgery only if complete cytoreduction was achieved; a small-volume residual disease at the end of surgery did not improve survival [30,31]. Neoadjuvant chemotherapy should not be offered in cases of tumours less sensitive to chemotherapy, such as low-grade serous and low-grade endometrioid carcinomas, clear-cell and mucinous carcinomas [24,25,26,28].

Ovarian cancer is predominantly a peritoneal disease, and therefore its non-resectability is based on the evaluation of abdominal sites critical for cytoreduction (disease location). The tumour size and FIGO stage also play a role in the decision-making process [30]. Lastly, objective quantification of tumour volume is suggested as a useful tool in the decision-making process. 

In many cases, non-resectability is related to the location more than the size of the metastases. The most clinically relevant disease sites predicting suboptimal cytoreduction have been reviewed in the literature (Table 1) and presented in 2019 by the European Society for Medical Oncology (ESMO)-ESGO consensus conference for ovarian cancer [31].

The ESMO-ESGO consensus conference listed nine markers of non-resectability, seven of which were in the abdomen (Figure 2), and one in the brain and one involving the lungs. Tumours were considered non-resectable if at least one of the critical sites for optimal cytoreduction was affected by the disease. A recent interim analysis of the multicentric prospective ISAAC (Imaging Study in Advanced OvArian Cancer) trial showed excellent performance of ESMO-ESGO non-resectability markers in preoperative imaging to triage candidates for upfront surgery or neoadjuvant chemotherapy [61].

Extra-abdominal non-resectable sites include brain or multiple lung metastases, as well as diffuse visceral pleural carcinomatosis. Brain metastases are rare at the initial diagnosis and are usually accompanied by symptoms such as headache, nausea or seizures. Lung metastases and upper mediastinal lymphadenopathy without infiltration of the lower mediastinal lymph nodes in the absence of retroperitoneal disease are also rare. The paracardiac (cardiophrenic) lymph nodes are associated with massive diaphragmatic carcinomatosis and are usually considered resectable [62]. For these reasons, the European Society of Urogenital Radiology (ESUR) suggests that routine preoperative staging should include assessment of the lower mediastinal and paracardiac lymph nodes and pleural effusion, in addition to the abdomino-pelvic scan [50,63]. If clinically indicated, the most extensive search for extra-abdominal metastases should be initiated. A study by Heitz et al. with 739 patients showed that the residual disease after incomplete cytoreductive surgery was mainly in the mesentery and serosa of the small bowel (79.8%); supradiaphragmatic metastases such as lung parenchymal and pleural metastases, mediastinal and supraclavicular lymph nodes (14.9%); porta hepatis/hepatoduodenal ligament (10.1%); liver parenchyma (4.3%); pancreas (8.0%); gastric serosa (3.2%); and coeliac trunk (2.7%) [64]. 

Another aspect that can be taken into consideration when planning surgery is the maximum size of metastases. A pooled analysis of EORTC-55971 and CHORUS in FIGO IIIC disease found a significantly better progression-free survival after cytoreduction in low-volume metastases (<5 cm) disease vs. large metastases (≥5 cm) disease; the latter group did not benefit from upfront cytoreduction when compared to neoadjuvant chemotherapy, and both approaches yielded the same overall survival [65]. Finally, this analysis has shown that patients with FIGO IV disease achieved a survival advantage when treated with neoadjuvant chemotherapy followed by interval debulking surgery [65]. However, this triage tool has not been universally accepted in the gynaecologic oncology community because some cases of FIGO IV disease (e.g., cardiophrenic lymph nodes, inguinal lymph nodes or port-site metastases after diagnostic laparoscopy) may behave more like FIGO III disease and should be treated accordingly. 

Several scoring systems have been proposed to standardize and improve the accuracy of this prediction, and the available predictive models for successful cytoreduction are discussed below. 

This narrative review aims to fill a gap in the literature on the diagnostic modalities used for predicting and assessing resectability in ovarian cancer.

## 2. Prediction of Non-Resectability

### 2.1. Laparotomy

Laparotomy is the most accurate way to evaluate the extent of disease and predict the surgical outcome. Some centres perform a limited open procedure rather than take the risk of denying surgery to a patient who potentially could have had a complete resection. In the Essen centre, two stop-or-go moments are suggested during surgery: (1) opening the lesser sac and assessing the pancreas, celiac trunk, liver, hepatoduodenal ligament, portal vein, hepatic artery and common bile duct, and (2) visualizing the radix mesenterii, superior mesenteric artery and small bowel after its mobilization and adhesiolysis [66]. 

Another trend towards more objective prediction of non-resectability is to quantify the tumour volume. For this purpose, the Peritoneal Cancer Index has been tested (Figure 3, Table 2). The Peritoneal Cancer Index, first proposed by Jacquet and Sugarbaker [67], is the most commonly used score to standardize the quantification of peritoneal spread in gastrointestinal cancer. Previous investigations have also demonstrated good performance of this surgical score in ovarian cancer, suggesting a good correlation with surgical and clinical outcome (Table 2) [68,69,70,71]. The Peritoneal Cancer Index is assessed by inspection of the peritoneum during laparotomy, and if the Peritoneal Cancer Index is high, the surgeon may choose not to proceed with immediate surgical cytoreduction [72]. The ability to predict non-resectability using the Peritoneal Cancer Index is shown in Table 2 with an area under the receiver operating curve (AUC) ranging from 0.69 to 0.94 [33,42,68,70,73,74,75,76,77,78]. High concordance for the Peritoneal Cancer Index scores from surgeons having variable experience is reported, indicating that the Peritoneal Cancer Index is a reproducible scoring index [71].

However, laparotomy is a highly invasive procedure for diagnosing non-resectability with a high risk of complications. Most importantly, it can delay the initiation of chemotherapy in cases of non-resectable disease and thereby jeopardise patient survival. To overcome this issue, the use of diagnostic laparoscopy or imaging techniques to predict non-resectability has been proposed, allowing better triage of patients for more individualized treatment.

### 2.2. Laparoscopy

Diagnostic laparoscopy before debulking surgery offers the following advantages: (a) assessment of resectability by direct and magnified visualization of the peritoneal cavity (Figure 4); (b) shorter operating time, faster recovery and earlier start of neoadjuvant chemotherapy when compared to laparotomy in case of non-resectable disease; and (c) collection of tissue for histopathologic assessments [94].

The assessment of critical sites for resectability and tumour extent on laparoscopy was already described by Vergote et al. in 1998 [95], but it was in 2005 that Fagotti et al. first published a prospective case series demonstrating an overall accuracy rate of 90% when comparing laparoscopy with laparotomy in the prediction of optimal cytoreduction (residuum ≤ 1 cm) [55]. However, the use of laparoscopy to predict resectability has some limitations, such as the evaluation of the lesser sac, behind the gastro-splenic ligament and the mesenteric root [96]. In 2013, the multicentric prospective OLYMPIA-MITO 13 study showed that the least assessable feature describing intraabdominal disease extent by laparoscopy was mesenteric retraction (only assessable in 25.8%, 31/120 patients) [96]. The remaining features were accurately assessed by almost all the centres during laparoscopy, ranging from 99.2% (peritoneal carcinomatosis) to 90% (bowel infiltration). In 2017, a multicentric randomized controlled trial in the Netherlands led by Rutten et al. evaluated non-resectability using predefined markers of non-resectability, such as (1) bowel serosal and/or mesenterial deposits, (2) non-resectable diaphragmatic peritoneal carcinomatosis and (3) extensive agglutinated intra-abdominal metastatic disease (including spleen and retrohepatic area). Using laparoscopy upfront improved optimal cytoreduction rates to 90% (92/102) vs. 61% (60/99) in patients who were randomized to the laparotomy without assessment by laparoscopy arm (relative risk 0.25; 95% CI, 0.13–0.47; *p* < 0.001) [45]. 

In order to objectively evaluate the tumour volume, in 2006 Fagotti et al. published the Fagotti score using a laparoscopic predictive index value to estimate the chances of having residual disease > 1 cm after cytoreduction. This scoring system is based on the presence of eight features: omental cake, peritoneal parietal carcinomatosis (except diaphragm), diaphragmatic carcinomatosis, mesenteric retraction, bowel and/or stomach infiltration and hepatic peritoneal implants. These parameters provide an indirect estimate of tumour load, and each parameter was assigned two points when present. A score ≥ 8 predicted residual disease with an overall accuracy of 75%, a positive predictive value of 100% and a negative predictive value of 70% [49]. In 2008, Fagotti et al. validated the performance of the model for predicting optimal cytoreduction (residual disease ≤ 1 cm) in a larger prospective study in advanced ovarian cancer [55]. For the purpose of minimizing the rate of inappropriate lack of exploration, i.e., patients with resectable disease that will not undergo upfront cytoreduction surgery, the overall laparoscopic score ≥ 8 was chosen, corresponding to a positive predictive value of 100%. However, using a laparoscopic score ≥ 8 led to unnecessary exploratory laparotomies (1-NPV [negative predictive score]) in 40.5% of the patients, in whom optimal cytoreduction was not possible to achieve [55]. Other authors used the same cut-off of 8 with a rate of unnecessary explorations (1-NPV) ranging from 4–71% and an AUC of 0.66–0.98 for predicting optimal cytoreduction (Table 2) [48,68,74,87,88,90,92,93]. Brun et al. [48] modified Fagotti score with a cut-off ≥ 4 using 4 out of 8 parameters. This simplified laparoscopy-based score was similarly accurate in predicting resectability (residuum ≤ 1cm), achieving a positive predictive value (PPV) of 100% and 57% rate of unnecessary explorations (1-NPV) (Table 2). In 2015, after the introduction of upper abdominal surgery performed by gynaecologic oncologists and maximal surgical effort to achieve no gross residual disease, the Fagotti score was updated to reflect these trends [94]. The newly suggested cut-off value of 10 offered better discrimination, with a lower rate of unnecessary laparotomies (33.2%) (Table 2). Furthermore, Fagotti excluded mesenterial retraction and carcinomatosis on the serosa of the small bowel from the scoring system, since these findings are now regarded as absolute criterions for non-resectability (Table 3) [39].

The Peritoneal Cancer Index in laparoscopy was used in 6 studies (Table 2) [73,78,87,89,91,92], with NPV ranging from 71 to 82%. Since there is no universal cut-off defined for the Peritoneal Cancer Index, several thresholds were used, varying between 10 and 20 (Table 2). Two of those studies also evaluated the prognostic performance of the Peritoneal Cancer Index in terms of overall survival and number of complications. Llueca et al. suggested that patients with a Peritoneal Cancer Index score >20 should undergo neoadjuvant chemotherapy in order to avoid high risk of complications from primary cytoreduction [78]. Climent et al. found a statistically significant correlation between reduced survival and high Peritoneal Cancer Index score (using score >10 and >20 as cut-offs) [92].

In 2019, a Cochrane Review evaluated the role of laparoscopy after conventional preoperative work-up for predicting residual disease after surgery in women with advanced ovarian cancer. It is of note that only two studies by Fagotti et al. used laparotomy as a reference standard in all the patients [35,86]. The NPV of laparoscopy ranged between 54–96% in 10 studies, of which 6 studies using the Fagotti score reported NPVs ranging from 75–100% [97]. Looking at the results using laparoscopy prior to laparotomy, this approach did not eliminate unnecessary surgical explorations, since for every 100 women referred for upfront debulking surgery after laparoscopy, 4–46 cases would be left with visible residual tumour. Moreover, a Cochrane review pointed out the bias in these studies (i.e., only two studies used laparotomy as reference standard) and concerns that routine implementation of laparoscopy in standard ovarian cancer workup would lead to many unnecessary exploratory laparotomies. In 2020, the ENGOT (European Network of Gynaecological Oncology Trial) group showed in their survey that laparoscopy was routinely performed to assess resectability in only 25.4% of European centres, and even in those, the treatment strategy was mostly not based on laparoscopic scores [42].

To avoid the risk that patients with ovarian cancer who are falsely assessed as non-resectable on preoperative imaging will miss a potential chance for optimal cytoreduction with a worsening prognosis, there are centres that use diagnostic laparoscopy with/without laparoscopic predictive scoring. In any case, laparoscopic assessment of resectability prior to laparotomy cannot prevent a significant number of subsequent unnecessary laparotomies.

### 2.3. Imaging

The ideal imaging modality for assessing non-resectability does not yet exist. The 2023 ESMO–ESGO-ESP Consensus Conference on Ovarian Cancer recommended abdominal contrast-enhanced (CE) computed tomography (CT), magnetic resonance imaging (MRI) or whole-body positron emission tomography (PET)-CT with the radiotracer [18F] Fluorodeoxyglucose (FDG) as viable options in the initial evaluation of patients with advanced ovarian cancer (Figure 4, Figure 5, Figure 6, Figure 7, Figure 8, Figure 9 and Figure 10). Transvaginal and transabdominal ultrasound by an expert sonographer may be used to assess tumour extent (Figure 4, Figure 5, Figure 6 and Figure 7) [31]. Advantages and disadvantages of the main modalities are compared in Table 4.

The reporting of preoperative imaging findings should provide a structured overview of likely disease extent and include detailed evaluation of all non-resectable sites. The non-resectable disease can be described either by positive critical site assessment or by using an objective score to predict non-resectability. Imaging alone cannot be used to determine the patient’s management [11]. Several scoring systems have been published with the aim to simplify and standardise preoperative staging and the decision whether to proceed with the primary surgery. The Peritoneal Cancer Index, adapted for imaging, is the only externally validated system (Table 2). The preoperative Peritoneal Cancer Index was studied using CT, whole-body diffusion-weighted imaging (WB-DWI)/MRI and PET/CT with surgical Peritoneal Cancer Index as the reference standard [33,73,78,84] (Table 2). The ongoing Imaging Study in Advanced ovArian Cancer (ISAAC) study is evaluating the diagnostic performance of Peritoneal Cancer Index using ultrasound, WB-DWI/MRI and CT (Clinicaltrials.gov: NCT03808792). The evaluation of the Peritoneal Cancer Index at the beginning of the surgery (laparotomy) was also investigated as a prognostic factor in ovarian cancer patients [68,69,77,98,99]. Several studies showed that a high CT-Peritoneal Cancer Index score is a predictor of surgical complications, lower disease-free survival and lower overall survival [84,99,100,101].

#### 2.3.1. Ultrasound

The recent European and international guidelines acknowledged that ultrasound imaging quality has improved in recent decades and, if carried out by an experienced sonographer, ultrasound has an invaluable role in estimating the malignant potential and histopathological features of ovarian tumours, as well as assessing tumour extent in the abdomen and pelvis [11,31]. Multiple prospective studies on large cohorts of patients showed good diagnostic performance in the assessment of ovarian cancer spread in the abdomen but also in the prediction of non-resectability [55,79,102,103,104]. Furthermore, ultrasound-guided tru-cut biopsy is possible in patients unfit for surgery or in whom secondary (metastatic) ovarian cancer is suspected (Table 4) [105,106,107]. Thoracic percutaneous ultrasound has limitations in visualizing mid-/upper mediastinal or lung parenchymal metastases, but convex array or linear array probes can visualize parietal pleural carcinomatosis and fluidothorax or cardiophrenic lymph nodes (anterior paracardiac lymph nodes (Figure 5 and Figure 6). Moreover, transabdominal ultrasound has a lower accuracy (compared to intraoperative findings) in the identification of small-size peritoneal implants [79,103]. To improve detection of small-volume carcinomatosis, laparoscopy prior to laparotomy was suggested as an addition to ultrasound examination in the preoperative assessment of ovarian cancer patients in cases with unclear resectability (for example, if bowel serosa and its mesentery seem uninvolved on preoperative imaging, but the loops are retracted to the mesentery with irregular dilatation and impaired peristalsis) [79,102].

To assess critical sites, Fischerova et al. evaluated the performance of transvaginal/transabdominal ultrasound, CT and WB-DWI/MRI to preoperatively assess ESMO-ESGO markers of non-resectability in the abdomen (Figure 2) [79]. All the three imaging methods yielded similar accuracy (85–90%), but ultrasound showed higher sensitivity than WB-DWI/MRI and CT (63%, 50% and 56%, respectively) for predicting non-resectability using ESMO-ESGO-specific markers. Ultrasound achieved the highest specificity, followed by WB-DWI/MRI and lastly CT for all evaluated markers of non-resectability (98%, 98% and 94%, respectively). The promising results from this single-unit study motivated the initiation of a prospective multicentric study (Imaging Study in Advanced ovArian Cancer (ISAAC trial), Clinicaltrials.gov: NCT03808792) comparing transvaginal/transabdominal ultrasound, CT and WB-DWI/MRI for predicting residual disease after surgery. The enrolment was completed in 10/2022 and the results are awaited in 2023. The results of an interim analysis showed that transvaginal/transabdominal ultrasound was non-inferior neither to CT (*p*-value = 0.029) nor to WB-DWI/MRI (*p*-value = 0.036). For predicting non-resectability, ultrasound yielded the best results with an AUC of 0.85, sensitivity of 91% and specificity of 86%. In comparison, CT and WB-DWI/MRI yielded lower AUCs (0.79 and 0.78, respectively) and sensitivities (89% and 87%), and CT yielded lower specificity than WB-DWI/MRI and ultrasound (69% vs. 80%/86%) (Figure 7) [61].

For the prediction of residual disease after surgery using predictive score, Testa et al. developed a predictive score for residual disease including: (1) peritoneal carcinomatosis, (2) bowel mesentery involvement, (3) omental involvement, (4) massive pelvic involvement and (5) ascites, awarding two points for every positive marker. Using the cut-off > 5 for residual disease, the sensitivity and specificity of the ultrasound score were 31% (20/64) and 92% (46/50), respectively, and PPV/NPV was 83%/51%. In order to prevent 17% (1-PPV) from being falsely subjected to no explorations, 39% (1-NPV) of the patients would undergo unnecessary laparotomies [55]. 

Ultrasound imaging has been accepted as a method of choice in the diagnosis of malignant ovarian tumours, but only trained operators can provide clinicians with systematic ovarian cancer staging and prediction of non-resectability, including ultrasound-guided tru-cut biopsy, during a single visit. Based on its availability, low cost, patient-friendly approach and reliability, the emphasis on formal training in gynaecologic oncology scanning as a part of gynaecologic oncology fellowship would be of benefit. Ultrasound cannot detect small-volume carcinomatosis, especially on bowel serosa, but in indeterminate cases can be combined with diagnostic laparoscopy.

#### 2.3.2. Computed Tomography

Abdominal CT has been recommended over the past few years as the modality of choice for staging ovarian cancer (Figure 7) [108]. To assess the peritoneal carcinomatosis, the sensitivity of CT ranges from 58% to 90%, specificity from 58% to 94% and accuracy from 76% to 90% (Table 5) [53,54,79,102,104,109]. CT is a widely available technique with the advantage of short examination time (Table 4). In addition, tru-cut biopsy may be carried out under computed tomography guidance, although ultrasound offers the advantage of dynamic image in real time, neither radiation load nor fasting prior to the procedure or the need for patients’ preparation (i.v. iodine contrast agent). CT-guided tru-cut biopsy will, however, be more beneficial at less accessible sites in the abdomen [73]. The disadvantage of CT is ionising radiation exposure and possible complications due to intravenous iodine contrast administration, mainly severe allergy or nephrotoxic effects with necessary dose reduction or no-contrast administration in patients with renal failure. For primary tumour characterization, CT has low soft-tissue resolution and is unsuited to differentiate between benign and malignant adnexal masses [110]. The drawback of CT for abdominal staging of ovarian cancer is its limitations in reliably visualising bowel surface and mesenterial cancer implants, and to differentiate parietal diaphragmatic from visceral liver carcinomatosis [50]. If CT with contrast cannot be carried out, or if CT findings are indeterminate, WB-DWI/MRI and/or PET/CT can be used, especially if retroperitoneal or supradiaphragmatic lymph node metastases are suspected [109] (Table 4).

Using abdominal and thoracic CT in the assessment of non-resectability based on critical sites was investigated in three studies [36,57,79]. In 1993, Nelson et al. described eight sites of disease: attachment of omentum to spleen (splenectomy was not performed at that time), lesion > 2 cm in mesentery, liver surface or parenchyma, diaphragm, gallbladder fossa, suprarenal para-aortic nodes, pericardiac nodes and pulmonary or pleural nodules with sensitivity/specificity of 92%/79% and PPV/NPV of 67%/96% [57]. In 2017, in the study of Michielsen et al., the metastatic sites considered non-resectable were duodenum, stomach, and celiac trunk carcinomatosis, diffuse serosal carcinomatosis, superior mesenteric artery, mesenteric root and suprarenal retroperitoneal lymphadenopathy. For these markers, the CT sensitivity/specificity was 66%/77% and PPV/NPV 77%/67% [36]. In 2022, Fischerova et al. used the ESMO-ESGO non-resectable criteria [31] for prediction of suboptimal cytoreduction of the disease-reporting sensitivity/specificity of 50%/98% and PPV/NPV of 89%/86% [79]. 

The role of preoperative CT in the assessment of non-resectability using prediction models has been explored by several authors (Table 2). Some have developed their original scores with AUCs ranging from 0.67–0.97 [40,43,44,46,47,51,58,80,81,82] and others have validated already published scores, such as the Peritoneal Cancer Index, with AUCs ranging from 0.55–0.76 [73,74,78,84]. Residual disease was frequently noted in patients with diffuse peritoneal thickening, mesenterial disease, suprarenal lymph nodes, ascites and disease on the diaphragm or liver (Table 1). These critical sites identified on CT were included in some scoring systems together with clinical features such as age, performance status by the Eastern Cooperative Oncology Group (ECOG PS) [115], classification of the American Society of Anesthesiologists physical status (ASA PS) [116] and serum tumour markers such as the Cancer antigen (CA) 125 [40,44,81,82]. Unfortunately, some of the models that report good performance for predicting residual disease failed external validation [61,82,117,118,119]; thus, external validation of their diagnostic performance before their integration into diagnostic algorithms is essential. 

CT remains a useful staging modality; it is widely available and has the advantage of rapid image acquisition. However, CT has several disadvantages, such as low soft-tissue resolution, which limits the ability to characterize primary tumours. CT also has limitations in detecting small-volume carcinomatosis, especially on the surface of the small bowel, and for detecting mesenterial cancer implants.

#### 2.3.3. Magnetic Resonance Imaging 

The addition of dynamic contrast-enhanced imaging and functional diffusion-weighted imaging to the morphological conventional MRI improves not only primary tumour characterization, but also the detection of peritoneal lesions [53,79,109,111], lymph node metastases and the prediction of residual disease after surgery [109,117,120,121,122] (Table 2 and Table 5). The advantage of MRI is its superior soft-tissue resolution, allowing excellent characterization of soft tissue. Diffusion-weighted imaging enables refined tumour characterization by depicting restricted motion of water molecules within hypercellular malignant tumours. Tissue diffusion is quantified by calculating apparent diffusion coefficient (ADC) values. The reason for the apparent high contrast between tumour tissue (with restricted diffusion) and normal tissue (with no restricted diffusion) is their different diffusion properties. Diffusion-weighted images are interpreted together with the ADC maps and morphological images. The application of sequences tailored for whole-body (WB) examinations makes DWI-MRI useful for tumour staging. Studies have shown similar performance of WB-DWI/MRI and positron emission tomography (PET)/CT for detecting retroperitoneal lymph node metastases and distant metastases, while WB-DWI/MRI yields better sensitivity than PET/CT and CT for peritoneal staging [123]. The main limitations for the use of DWI/MRI as a first-line modality for tumour characterisation and staging are high cost, long examination time, occasionally inadequate images due to organ movements (e.g., respiratory, intestinal peristalsis) and limited availability, but also limited evidence on its accuracy and role in ovarian cancer staging (Table 4) [108,124]. 

To predict non-resectability based on the evaluation of critical sites (as presented in the previous section on CT), Michielsen et al. (2017) showed higher sensitivity (94%), specificity (98%) and accuracy (96%) of WB-DWI/MRI in the detection of disease sites indicating non-resectability when compared to CT (sensitivity 66%, specificity 77% and accuracy 71%) [36]. Fischerova et al., using ESMO-ESGO criteria of non-resectability, did not show statistically different results between WB-DWI/MRI (AUC 0.75), pelvic and abdominal ultrasound (0.80) and contrast-enhanced CT (AUC 0.74) for prediction of residual disease at the end of surgery [79].

In addition to the assessment of involved critical sites, WB-DWI/MRI performance was assessed through the calculation of predictive scores. In 2019, Engbersen et al. presented good results in the prediction of surgical outcome (AUC = 0.98) using the Peritoneal Cancer Index with an excellent inter-observer agreement and intraclass correlation coefficient of 0.90 (95% CI: 0.64–0.96) [33]. These results are in agreement with those of Rizzo et al. published in 2020. The authors demonstrated that a nomogram using WB-DWI/MRI was accurate and had better sensitivity than CT for the assessment of multiple sites of disease in epithelial ovarian cancer. They also showed that WB-DWI/MRI was significantly more accurate than CT to detect disease involving some unresectable sites, such as mesenteric carcinomatosis and large bowel carcinomatosis (Table 2) [53]. 

WB-DWI/MRI can characterize primary tumours but also precisely detect tumour spread and predict non-resectability. Some promising data based on few studies show higher sensitivity of WB-DWI/MRI than CECT, PET/CT or ultrasound for diagnosing small-volume carcinomatosis, especially on the bowel serosa, but more research is needed before introducing this modality in standardised staging pathways. It also has limitations such as long examination time, limited availability, contraindications and need of expertise. 

#### 2.3.4. Positron Emission Tomography 

Positron emission tomography uses positron-emitting radiolabelled molecules to display molecular interactions of biological processes in vivo. The most used radioisotope tracer is 18F-fluoro-deoxyglucose (FDG), a glucose analogue that is preferentially taken up by and retained by malignant cells [125]. The uniquely high sensitivity of PET—in the picomolar range—may allow detection of even minute amounts of radiolabelled markers in vivo, making PET the modality of choice for molecular imaging. However, PET is a functional imaging method without anatomic correlations, and for this reason it needs to be combined with morphological imaging such as CT and/or MRI [118]. 

PET/CT combines the anatomic details depicted with CT and metabolic information obtained with PET, yielding more precise anatomic information, and reducing the equivocal PET interpretations (Figure 9 and Figure 10). PET/CT has disadvantages such as radiation exposure, high cost, limited spatial resolution and limited depiction of small volumes of metabolically active tumours (Table 4) [119]. For characterizing complex adnexal masses owing to the high rate of false positive and false negative findings, PET/CT is of no advantage [117,126]. PET/CT is commonly used in staging of ovarian cancer as a problem-solving modality in indeterminate supra-diaphragmatic lesions and retroperitoneal lymph node involvement (Table 4, Figure 10) [54,109,112,113,114,127,128,129].

To predict non-resectability based on the critical sites, Alessi et al. focused their preoperative analysis on the evaluation of the hepatic hilum, root mesentery with retraction and involvement of pancreas and duodenum. In 6/21 (29%) cases, the PET/CT showed hepatic hilum infiltration (*n* = 4) and mesentery involvement (*n* = 2) that was confirmed at surgical exploration and excluded optimal cytoreduction. The PET/CT did not find any limiting factors in the remaining 15/21 (71%) patients in whom optimal cytoreduction was achieved [32].

The use of prediction models based on PET/CT in the assessment of non-resectability was described in 2015 by Shim et al., who developed and validated a nomogram based on a surgical aggressiveness index and five PET/CT features: diaphragm disease, ascites, peritoneal carcinomatosis, small bowel mesentery implant and tumoural uptake ratio, which is the relation between the highest maximum standard uptake value (SUVmax) in the upper abdominal region/lower abdominal region. The accuracy of the nomogram was very good in both development and validation cohorts (concordance index = 0.88 and 0.86, respectively) [59]. In 2019, Chong et al. also developed a scoring system for predicting suboptimal cytoreduction. They included the ECOG PS and another four metabolic parameters assessed by PET/CT: SUVmax of central (OR, 5.250), right upper (OR, 4.148) and left upper (OR, 5.921) regions of the abdomino-pelvic cavity similar to the division of the Peritoneal Cancer Index score [67] and lymph node regions (OR, 4.148). The latter metabolic parameter (lymph node regions) was calculated as the sum of the SUVmax values for the three lymph node regions (pelvis, para-aortic and extra-abdominal). These five parameters were associated with suboptimal cytoreduction (AUC = 0.78). However, external validation of the Peritoneal Cancer Index only reached an AUC of 0.58 [85]. In 2020, Gu et al. defined a scoring system based on the eight radiological criteria of the Suidan score, which was originally developed for CT assessment [40]. These radiological criteria comprised lesions in splenic hilum/ligaments, gastrohepatic ligament/porta hepatis, retroperitoneal lymph nodes above the renal hilum (including supradiaphragmatic), diffuse small bowel adhesions/thickening, abdominal ascites (moderate-severe), gallbladder fossa/liver intersegmental fissure lesion, lesser sac lesion >1 cm and root of the superior mesenteric artery lesion (Table 2). This scoring system using PET/CT predicted complete resection with an AUC of 0.80 [75].

PET combined with MRI (PET/MRI) has recently been introduced and causes less radiation exposure than PET/CT, and MRI provides better soft tissue resolution than CT. Further investigations are needed to clarify the role of PET/MRI in patients with ovarian cancer [130].

PET/CT is not beneficial for characterizing primary ovarian tumours but may be used for staging as a problem-solving tool if unclear CT findings, such as indeterminate lymph node involvement in the retroperitoneum or mediastinum, are detected. PET/CT is unable to detect small-volume carcinomatosis, especially on the bowel serosa and its mesentery or on liver surface. 

## 3. Future Studies

Ongoing studies explore the timing of ovarian cancer surgery, compare the diagnostic performance of the different imaging modalities and assess the added value by incorporating radiomic tumour features. Furthermore, combining information from advanced imaging markers and the preoperative biopsy, e.g., tumour histotypes and tumour mutational burden, may serve as tools to stratify patients for more individualized treatment.

The completion of the ongoing clinical trials, such as the multicentric ISAAC trial (imaging) and TRUST-AGO-OVAR-OP.7 (surgical), will provide new data to help triage patients for either upfront debulking surgery or neoadjuvant chemotherapy.

## 4. Conclusions

Correct selection of patients for debulking surgery is pivotal in managing patients with advanced ovarian cancer. The final decision for type of treatment is, however, a multidisciplinary decision involving experienced gynaecologic onco-surgeons and radiologists, and should be based on a combination of the patient’s overall clinical picture, symptoms, personal preferences, previous medical and surgical history and biomarkers (radiological, genetic, immunological). No single diagnostic modality should determine the patient’s journey. Until recently, CT was the imaging method of choice for staging of ovarian cancer and predicting resectability. Nevertheless, the development of new techniques, such as WB-DWI/MRI and PET/CT, and the constant improvement in ultrasound imaging, has led to their incorporation in preoperative work-up at many centres. In general, the above imaging modalities have high specificity for predicting residual disease, but low sensitivity for detecting small-volume carcinomatosis, leading to unnecessary surgical explorations. Laparoscopy can be used as a second-stage test in cases of uncertain resectability to exclude small-volume carcinomatosis on bowel and its mesentery. Various models and scoring systems have been proposed with the aim of predicting surgical outcome, but their good performance was not reproduced at external validation. More accurate prediction of residual disease may be available in the future, but meanwhile a careful imaging assessment of sites critical for ovarian cancer surgery remains the most useful approach. 

## Figures and Tables

**Figure 1 cancers-15-01904-f001:**
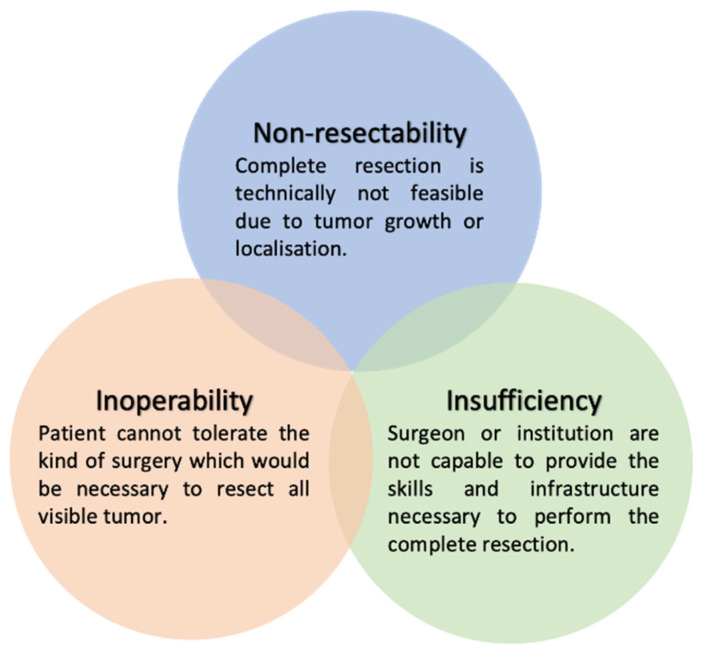
Reasons for suboptimal surgical outcome.

**Figure 2 cancers-15-01904-f002:**
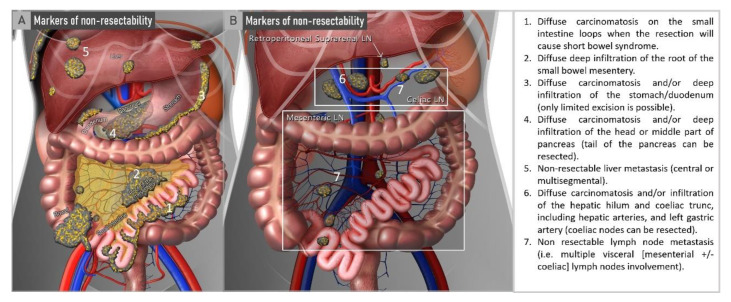
ESMO-ESGO markers of non-resectability. Non-resectable disease is defined by one or more markers published by ESMO-ESGO [31]. ESGO, European Society of Gynaecological Oncology; ESMO, European Society for Medical Oncology.

**Figure 3 cancers-15-01904-f003:**
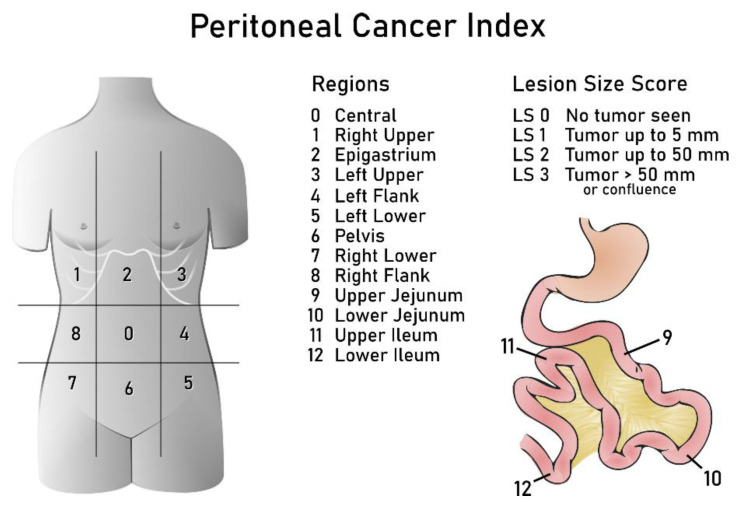
Peritoneal cancer index.

**Figure 4 cancers-15-01904-f004:**
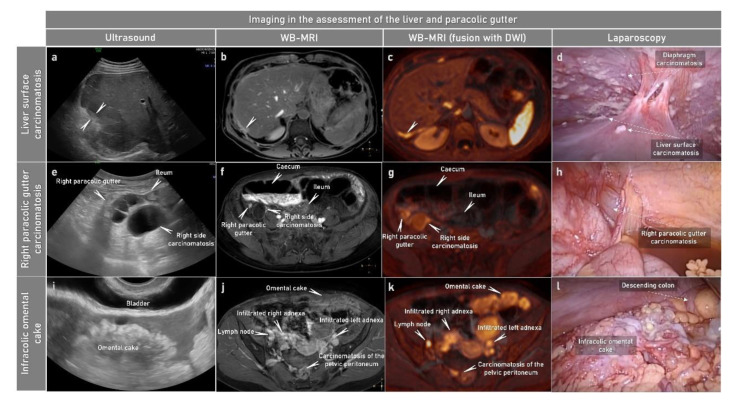
Characteristic imaging and laparoscopy findings in patient with high-grade serous cancer FIGO stage IVB. Abdominal convex array ultrasound (first column), axial WB-MRI: CE-T1WI-FS (contrast enhanced T1 weighted imaging with fat suppression) (second column) and WB MRI fused with DWI: DWIBS (diffusion-weighted imaging with background body signal suppression) + CE-T1WI-FS (third column) and laparoscopy findings (fourth column). The imaging findings confirmed by laparoscopy indicate visceral hepatic carcinomatosis marked with arrow outline (**a**–**d**), mainly cystic carcinomatosis in the lower part of right paracolic gutter (**e**–**h**) and omental cake (**i**–**l**). CE, contrast-enhanced; DWI, diffusion-weighted imaging; FIGO, International Federation of Gynecology and Obstetrics; FS, fat suppression; WB-MRI, whole-body magnetic resonance imaging.

**Figure 5 cancers-15-01904-f005:**
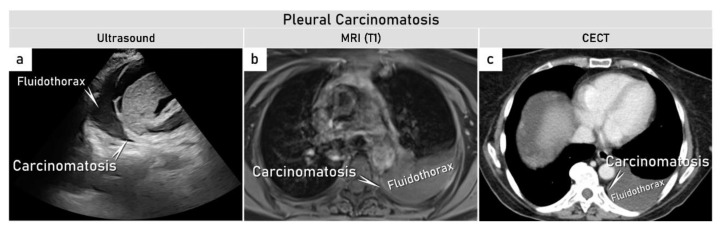
Demonstration of a patient with ovarian cancer (FIGO stage IVB) with pleural parietal carcinomatosis. Sagittal ultrasound images showing left parietal pleural carcinomatosis and hydrothorax (**a**), with corresponding axial WB-MRI (CE-T1WI-FS) (**b**) and CECT (**c**). CECT, contrast enhanced computed tomography; CE-T1WI-FS, contrast-enhanced T1-weighted imaging with fat suppression; FIGO, International Federation of Gynecology and Obstetrics; WB-MRI, whole-body magnetic resonance imaging.

**Figure 6 cancers-15-01904-f006:**
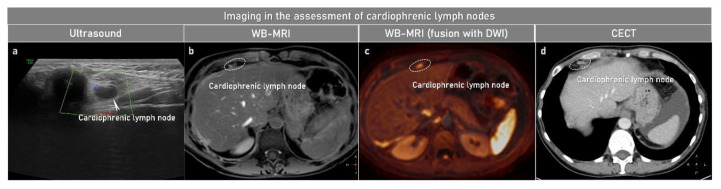
Cardiophrenic lymph node assessment using different imaging methods in high-grade serous cancer stage IVB. Ultrasound linear array probe (first column), axial WB-MRI: CE-T1WI-FS (contrast-enhanced T1-weighted imaging with fat suppression) (second column), WB-MRI (fusion with DWI): Fusion of DWIBS (diffusion-weighted imaging with background body signal suppression) + CE-T1WI-FS (third column) and CECT (fourth column) are demonstrating metastatic cardiophrenic lymph node of 7 mm marked with arrow on ultrasound (**a**) and circle on MRI (**b**,**c**) and CECT (**d**). This is the same case as presented in Figure 4. CE, contrast-enhanced; CT, computed tomography; DWI, diffusion-weighted imaging; FS, fat suppression; WB-MRI, whole-body magnetic resonance imaging.

**Figure 7 cancers-15-01904-f007:**
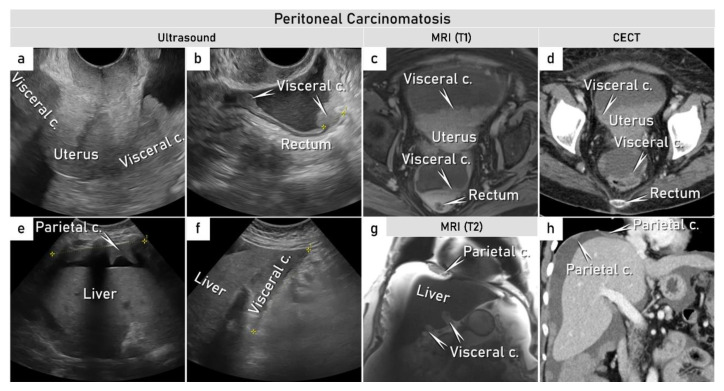
Patient with ovarian cancer (FIGO stage IVB) with pelvic and abdominal peritoneal carcinomatosis depicted by transvaginal and transabdominal ultrasound, MRI and CT at primary diagnostic work-up. Transvaginal sagittal ultrasound depicting hypoechogenic visceral carcinomatosis infiltrating the bladder and uterovesical fold ventral of the uterus. Dorsal of the uterus there is carcinomatosis in the rectosigmoid pouch and in the pouch of Douglas (**a**). Sagittal ultrasound depicts visceral carcinomatosis infiltrating the hypoechogenic muscle layer of the rectosigmoid, maximum infiltration length is marked with yellow calipers (**b**). Axial CE-T1WI-FS, contrast-enhanced T1-weighted imaging with fat suppression (**c**) and contrast-enhanced computed tomography (CECT) (**d**). Transabdominal ultrasound depicts parietal diffuse carcinomatosis on the diaphragm (**e**) and visceral part of liver surface (**f**) with corresponding findings at coronal T2-weighted MRI (**g**) and CECT (coronal reconstruction) (**h**). Maximum length of carcinomatosis is marked with yellow calipers (**e**,**f**). CECT, contrast-enhanced computed tomography; CE-T1WI-FS, contrast-enhanced T1 weighted imaging with fat suppression; FIGO, International Federation of Gynecology and Obstetrics; T2 MRI, T2-weighted magnetic resonance. This is the same case as presented in Figure 5.

**Figure 8 cancers-15-01904-f008:**
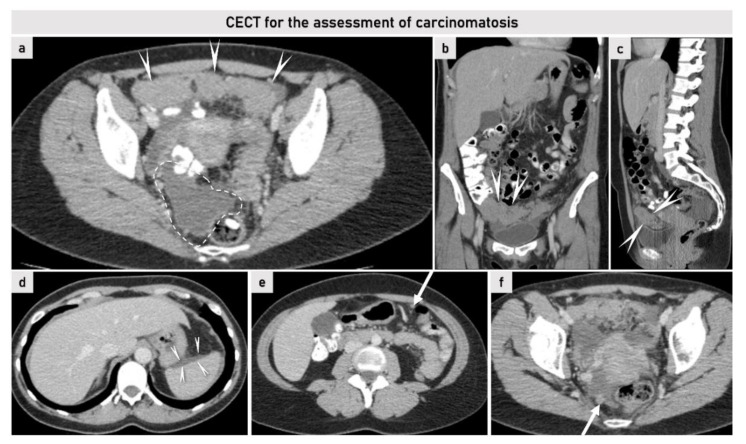
CT at primary diagnostic work-up in a patient aged 45 years presenting with symptoms of bloating and obstipation. Contrast-enhanced CT shows large omental mass ventral to the uterus and cranial to the bladder ((**a**–**c**); marked with arrow outlines). Pelvic peritoneal carcinomatosis is indicated by a dotted line (**a**) and arrow (**f**). Peritoneal lesions in the upper abdomen can be seen on the surface of the spleen marked with an arrow outlines (**d**), as well as nodular infiltration of infracolic omentum marked with an arrow (**e**). The patient was diagnosed with ovarian cancer (adenocarcinoma) FIGO stage IIIC. Figures (**a**,**d**–**f**), axial (transverse) plane; figure (**b**), coronal plane; figure (**c**), sagittal plane. CECT, contrast-enhanced computed tomography; FIGO, International Federation of Gynecology and Obstetrics.

**Figure 9 cancers-15-01904-f009:**
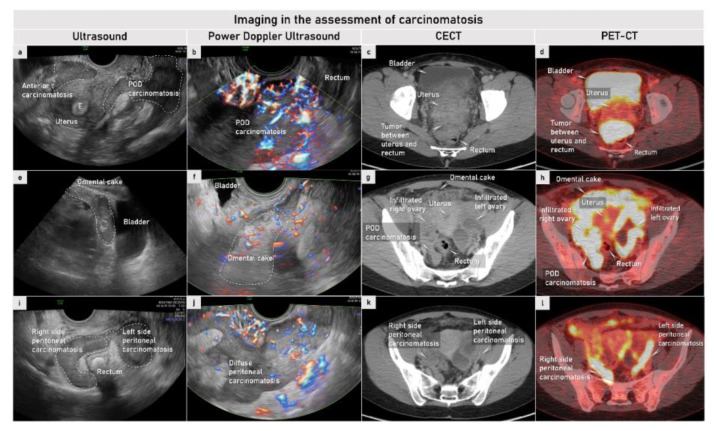
Advanced high-grade serous tubal cancer FIGO stage IVB. Sagittal plane of transvaginal ultrasound image demonstrating hypoechogenic diffuse carcinomatosis infiltrating the anterior and posterior compartments (**a**); carcinomatosis is highly perfused on Power Doppler ultrasound (**b**); axial (transverse) plane in CECT showing peritoneal carcinomatosis as a solid enhanced tumour between uterus and rectum and PET/CT demonstrating FDG avid peritoneal carcinomatosis tissue (**c**,**d**). Diffuse infiltration of infracolic omentum (omental non-homogenous cake) demonstrated on sagittal plane on gray-scale transabdominal scan with moderate perfusion detected by Power Doppler on transvaginal ultrasound (**e**,**f**); corresponding images from axial (transverse) plane in CECT (increased density in the omental fatty tissue) and PET/CT (FDG avid infiltration of omentum) (**g**,**h**). Diffuse hypoechogenic pelvic carcinomatosis on pelvic side walls in transverse plane in gray-scale transvaginal scan with moderate perfusion demonstrated with Power Doppler (**i**,**j**), and with corresponding images from CECT (thickening and enhancement of peritoneal reflection) and PET/CT (FDG avid peritoneal carcinomatosis tissue) in axial (transverse) plane (**k**,**l**). CECT, contrast-enhanced computed tomography; E, secretory endometrium; FIGO, International Federation of Gynecology and Obstetrics; PET/CT, positron emission tomography/computed tomography; FDG, fluorodeoxyglucose; POD, pouch of Douglas.

**Figure 10 cancers-15-01904-f010:**
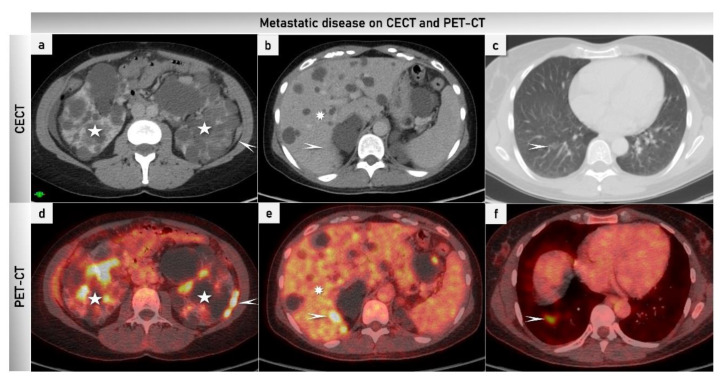
High-grade serous tubal cancer FIGO stage IVB. Axial CECT and PET-CT scans of the same patient as presented in Figure 9 are demonstrating the left paracolic peritoneal carcinomatosis implant (marked with arrows) and polycystic kidneys bilaterally (stars) (**a**,**d**); the carcinomatosis has high metabolic activity on PET/CT, within the renal calyces the excretion of FDG is visible (**d**). The liver shows multiple cysts (marked with a star; the arrow points to a nodule of visceral carcinomatosis on the liver surface with metabolic activity in the PET-CT; however, it is barely visible on CT (**b**,**e**)). A metastatic nodule of the right lung (marked with the arrow) with metabolic activity on PET-CT and minimal detectability on CT (ground glass appearance) (**c**,**f**). CECT, contrast-enhanced computed tomography; FDG, fluorodeoxyglucose; PET/CT, positron emission tomography/computed tomography.

**Table 1 cancers-15-01904-t001:** ESMO-ESGO markers of non-resectability paired with the data from the literature.

ESMO-ESGO Markers of Non-Resectability	References (*n* = 29)	% (*n*)
Diffuse carcinomatosis of the small bowel involving such large parts that resection would lead to short bowel syndrome (remaining bowel < 1.5 m)	[32,33,34,35,36,37,38,39,40,41,42,43,44,45]	48 (14/29)
Diffuse deep infiltration of the root of small bowel mesentery	[32,33,34,35,36,37,38,39,40,41,42,44,45,46,47,48,49,50,51,52,53,54,55,56,57,58,59,60]	97 (28/29)
Diffuse involvement/deep infiltration of: Stomach/duodenum; Head or middle part of pancreas.	[32,33,36,39,41,42,48,49,51]	31 (9/29)
Central or multisegmental parenchymal liver metastases	[32,33,34,38,39,41,42,48,49,50,51,57,60]	45 (13/29)
Involvement of coeliac trunk, hepatic arteries or left gastric artery	[32,34,36,37,38,40,41,47,50,51,57,60]	41 (12/29)
Non-resectable lymph node metastases	[34,36,38,40,41,42,47,50,51,52,56,57,58]	45 (13/29)
Multiple parenchymal lung metastases (preferably histologically proven)	[35,36,41,43,46,57]	17 (5/29)
Brain metastases	[35,36]	7 (2/29)

**Table 2 cancers-15-01904-t002:** Prediction of non-resectability based on preoperative imaging methods and laparoscopy.

	Date	Study Type	Patients (*n*)	Type of Model	Imaging Modality (Cut-Off ^1^)	Sensitivity (%)	Specificity (%)	PPV (%)	NPV (%)	Accuracy (%)	AUC	Outcome
ULTRASOUND												
Testa et al. [55]	2012	Prospective	147	Scoring system	>5	31	92	83	51	58	-	>1 cm residual disease
Fischerova et al. [79] *	2022	Prospective	67	Multivariable analysis	-	63	98	91	89	90	0.80	>1 cm residual disease
CT												
Nelson et al. [57]	1993	Retrospective	42	Multivariable analysis	-	92	79	67	96	-	-	≥2 cm residual disease
Bristow et al. [47]	2000	Retrospective	41	Scoring system	≥4	100	85	88	100	93	0.97	>1 cm residual disease
Dowdy et al. [43]	2004	Retrospective	87	Multivariable logistic regression	-	52	90	68	82	79	-	>1 cm residual disease
Axtell et al. [80]	2007	Retrospective	65	Multivariable logistic regression	-	79	75	46	93	77	-	>1 cm residual disease
Axtell et al. [80]	2007	Retrospective	87	External validation Axtell et al.	-	72	56	48	78	64	-	>1 cm residual disease
Ferrandina et al. [56]	2009	Prospective	195	Scoring system	-	24	98	93	50	56	0.82	>1 cm residual disease
Gerestein et al. [81]	2011	Multicentric prospective	115	Nomogram	-	-	-	-	-	74	0.67	>1 cm residual disease
Suidan et al. [44]	2014	Multicentric prospective	350	Scoring system	-	-	-	-	-	-	0.76	>1 cm residual disease
Janco et al. [82]	2015	Retrospective	279	Nomogram	-	-	-	-	-	-	0.75	Any visible disease
Borley et al. [46]	2015	Retrospective	111	Scoring system	-	69	71	75	65	-	0.75	>1 cm residual disease
Borley et al. [46]	2015	Retrospective	70	External validation Borley et al.	-	65	68	-	-	-	0.72	>1 cm residual disease
Son et al. [58]	2016	Retrospective	220	Scoring system	-	71	74	-	-	-	0.79	>1 cm residual disease
Son et al. [58]	2016	Prospective	107	External validation Son et al.	-	69	73	-	-	-	0.76	>1 cm residual disease
Suidan et al. [40]	2017	Multicentric prospective	350	Scoring system (same population of Suidan 2014 [44])	≥3	68	76	68	76	72	0.72	Any visible disease
Michielsen et al. [36] *	2017	Prospective	161	Multivariable analysis	-	66	77	77	67	71	0.72	Any visible disease
Feng et al. [74]	2018	Prospective	100	External validation Suidan et al. [40]	≥3	-	-	-	-	-	0.55	Any visible disease
Llueca et al. [78]	2018	Retrospective	49	External validation PCI score	>20	27	91	33	89	-	-	>1 cm residual disease
Fuso et al. [51]	2019	Retrospective	61	Scoring system	>8	85	100	100	60	92	0.95	Any visible disease
Ahmed et al. [73]	2019	Prospective	80	External validation PCI score	<20	90	39	75	70	69	-	≥1 cm residual disease
Kumar et al. [83]	2019	Retrospective	276	External validation Suidan et al. [44]	-	-	-	-	-	-	0.65	>1 cm residual disease
Kumar et al. [83]	2019	Retrospective	276	External validation Suidan et al. [40]	-	-	-	-	-	-	0.76	Any visible disease
Avesani et al. [84]	2020	Retrospective	297	External validation PCI score	-	-	-	-	-	-	0.64	Any visible di-sease
Fischerova et al. [79] *	2022	Prospective	67	Multivariable analysis	-	56	94	75	87	85	0.75	>1 cm residual disease
WB-DWI/MRI												
Michielsen et al. [36] *	2017	Prospective	161	Multivariableanalysis	-	94	98	98	94	96	0.96	Any visible disease
Engbersen et al. [33]	2019	Prospective	25	External validation PCI score	<15	100	88	-	-	-	0.98	Any visible di-sease
Rizzo et al. [53]	2020	Prospective	92	Nomogram	-	-	-	-	-	-	0.88	>1 cm residual disease
Fischerova et al. [79] *	2022	Prospective	67	Multivariableanalysis	-	50	98	89	86	87	0.74	>1 cm residual disease
PET/CT												
Shim et al. [59]	2015	Retrospective	240	Nomogram	-	66	88	-	-	-	0.88	Any visible disease
Shim et al. [59]	2015	Retrospective	103	External validation Shim et al.	-	-	-	-	-	-	0.86	Any visible disease
Alessi et al. [32]	2016	Prospective	23	Multivariable analysis	-	100	100	-	-	-	-	Any visible disease
Chong et al. [85]	2019	Retrospective	51	Scoring system	>10	82	65	-	-	-	0.78	>1 cm residual disease
Chong et al. [85]	2019	Retrospective	51	External validation PCI score	-	-	-	-	-	-	0.56	>1 cm residual disease
Gu et al. [75]	2020	Prospective	31	External validation Suidan et al. [40]	-	-	-	-	-	-	0.80	Any visible disease
LAPAROSCOPY												
Fagotti et al. [49]	2006	Prospective	64	Fagottic score	≥8	30	100	70	100	75	-	>1 cm residual disease
Fagotti et al. [86]	2008	Prospective	113	External validation Fagotti score	≥8	70	100	100	60	-	-	>1 cm residual disease
Brun et al. [48]	2008	Retrospective	55	External validation Fagotti score	≥8	46	89	89	44	60	0.74	>1 cm residual disease
Brun et al. [48]	2008	Retrospective	55	Scoring system	≥4	35	100	100	43	56	0.68	>1 cm residual disease
Chéreau et al. [68]	2010	Retrospective	61	External validation Fagotti score	<8	-	-	-	-	-	0.66	Any visible disease
Chéreau et al. [68]	2010	Retrospective	61	External validation Brun et al. [48]	<4	-	-	-	-	-	0.76	Any visible disease
Varnoux et al. [87]	2013	Prospective	29	Multivariable analysis	-	100	40	61	100	-	0.70	Any visible disease
Varnoux et al. [87]	2013	Prospective	29	External validation Brun et al. [48]	≥4	100	47	64	100	73	-	Any visible disease
Varnoux et al. [87]	2013	Prospective	29	External validation Fagotti score	≥8	100	47	64	100	73	-	Any visible disease
Varnoux et al. [87]	2013	Prospective	29	External validation PCI score	≥10	64	93	90	74	79	-	Any visible disease
Petrillo et al. [39]	2015	Prospective	135	Scoring system	≥10	47	97	100	67	-	0.89	>1 cm residual disease
Rutten et al. [45]	2017	Multicentric prospective	63	Multivariable analysis	-	-	-	-	84	84	-	>1 cm residual disease
Tomar et al. [88]	2017	Prospective	73	External validation Fagotti score	≥8	85	100	100	96	97	0.98	>1 cm residual disease
Feng et al. [74]	2018	Prospective	39	External validation Fagotti score	<8	-	-	-	-	-	0.71	Any visible disease
Ghisoni et al. [89]	2018	Multicentre retrospective	65	External validation PCI score	>16	63	90	71	86	82	-	Any visible disease
Hansen et al. [90]	2018	Prospective	226	External validation Fagotti score	≥8	71	49	85	29	67	-	Any visible disease
Llueca et al. [78]	2018	Retrospective	80	External validation PCI score	>20	38	88	33	90	-	-	>1 cm residual disease
Ahmed et al. [73]	2019	Prospective	80	External validation PCI score	<20	89	42	76	71	71	-	≥1 cm residual disease
Angeles et al. [91]	2021	Retrospective	43	External validation PCI score	-	-	-	-	-	-	0.90	Any visible disease
Climent et al. [92]	2021	Retrospective	34	External validation Fagotti score	≥8	14	81	16	78	68	0.66	>1 cm residual disease
Climent et al. [92]	2021	Retrospective	34	External validation PCI score	≥20	43	88	50	78	79	-	>1 cm residual disease
Llueca et al. [93]	2021	Retrospective	103	External validation Fagotti score	<4	86	74	-	-	-	0.83	>1 cm residual disease
LAPAROTOMY												
Chéreau et al. [68]	2010	Prospective	61	External validation PCI score	<10	-	-	-	-	-	0.69	Any visible disease
Espada et al. [34]	2013	Prospective	34	Scoring system	≥4	88	89	70	96	88	0.95	>1 cm residual disease
Lampe et al. [70]	2015	Retrospective	98	External validation PCI score	-	-	-	-	-	-	0.84	Any visible disease
Kasper et al. [60]	2016	Prospective	99	Scoring system	≥14	70	94	83	88	-	91	>1 cm residual disease
LLueca et al. [78]	2018	Retrospective	80	External validation PCI score	>20	73	81	38	95	-	-	>1 cm residual disease
Rosendahl et al. [77]	2018	Prospective	507	External validation PCI score	-	-	-	-	-	-	0.75	Any visible disease
Rosendahl et al. [77]	2018	Prospective	507	Score (PCI-2 + 9–12)	4	78	70	-	-	-	0.79	Any visible disease
Ahmed et al. [73]	2019	Prospective	80	External validation PCI score	<20	91	83	88	90	89	-	≥1 cm residual disease
Engbersen et al. [33]	2019	Prospective	25	External validation PCI score	-	-	-	-	-	-	0.92	Any visible disease
Feng et al. [74]	2018	Prospective	109	External validation PCI score	-	-	-	-	-	-	0.80	Any visible disease
Gu et al. [75]	2020	Prospective	31	External validation PCI score	-	-	-	-	-	-	0.81	Any visible disease
Zhou et al. [42]	2020	Retrospective	400	Scoring system	-	-	-	-	-	-	0.75	>1 cm residual disease
Zhou et al. [42]	2020	Retrospective	400	External validation PCI score	-	-	-	-	-	-	0.79	>1 cm residual disease
Zhou et al. [42]	2020	Retrospective	400	External validation Petrillo et al. [39]	-	-	-	-	-	-	0.74	>1 cm residual disease
Jónsdóttir et al. [76]	2021	Prospective	167	External validation PCI score	≥24	-	-	-	-	-	0.94	Any visible disease

^1^ Score cut-off value for prediction of suboptimal cytoreduction; AUC, area under the curve; CT, computed tomography; NPV, negative predictive value; PIV, predictive index value; PPV, positive predictive value; WB-DWI/MRI, whole-body diffusion-weighted magnetic resonance imaging. * Studies that used the same cohort of patients to evaluate the different imaging methods.

**Table 3 cancers-15-01904-t003:** Updated Fagotti score [39].

Parameters	Score 2 If:
Omental disease	Tumour infiltration of the greater omentum up to the large curvature of the stomach (infiltration of supracolic omentum)
Liver metastases	Any surface lesion larger than 2 cm
Lesser omentum and/or stomach and/or spleen involvement	Presence of obvious neoplastic involvement of the stomach and/or lesser omentum and/or spleen
Parietal peritoneal carcinomatosis	Massive peritoneal involvement and/or a miliaric pattern of distribution for parietal peritoneal carcinomatosis
Diaphragmatic disease	Widespread infiltrating carcinomatosis and/or confluent nodules to the most part of the diaphragmatic surface
Bowel infiltration	Large/small bowel infiltration (excluding recto-sigmoid involvement) *

* Rectosigmoid infiltration is not included in Fagotti score, due to its pelvic localization and given that the posterior exenteration is considered a standard surgical procedure in advanced epithelial ovarian cancer.

**Table 4 cancers-15-01904-t004:** Comparison of different diagnostic imaging methods used in ovarian cancer.

	Transvaginal and Transabdominal US	CE-CT	Whole-Body Diffusion-Weighted Imaging (DWI)/MRI	PET-CT
Advantages	Low cost High availability Exam duration~15–20 min Dynamic examination No radiation exposure No patient preparation No contraindications Ultrasound-guided tru-cut biopsy	High availability Exam duration < 5 min No patient preparation CT-guided tru-cut biopsy of less accessible abdominal sites	Detection of small-volume disease (bowel serosa and mesentery) Differentiation of distant metastases and metastatic retroperitoneal-and supradiaphragmatic lymph nodes from benign processes No radiation exposure	Differentiation of distant metastases and metastatic retroperitoneal and supradiaphragmatic lymph nodes from benign processes
Disadvantages	Limited visualization of chest and bones Insufficient detection of small-volume disease (bowel serosa and mesentery) Low image quality for retroperitoneum in obese patients	Radiation exposure Insufficient detection of small-volume disease (bowel serosa and mesentery) Iodine-based contrast: Contraindicated if previous severe allergy to contrast	Low availability Low experience in acquisition and interpretation High cost Antiperistaltic agent Exam duration > 45 min MRI-guided tru-cut biopsy limited by cost and availability of non-magnetic biopsy equipment Contraindicated by non-MRI-conditional implants, cardiac pacemaker, cochlear implants or severe claustrophobia Gd-based contrast: contraindicated if previous severe allergy to contrast	High cost Radiation exposure Exam duration ~30–40 min Insufficient detection of small-volume disease (bowel serosa and mesentery)

CE—contrast-enhanced; CT—computed tomography; Gd—gadolinium; MRI—magnetic resonance imaging; PET—positron emission tomography; US—ultrasound.

**Table 5 cancers-15-01904-t005:** Diagnostic accuracy of preoperative imaging techniques for detecting overall peritoneal carcinomatosis.

	Date	Study Type	Patients (*n*)	Sensitivity (%)	Specificity (%)	PPV (%)	NPV (%)	Accuracy (%)
ULTRASOUND								
Tempany et al. [104] *	2000	Multicentric prospective	280	61	95	61	95	91
Testa et al. [55]	2012	Prospective	147	90	96	94	92	93
Fischerova et al. [103]	2017	Prospective	394	70	98	89	93	92
Alcázar et al. [102] *	2019	Prospective	93	70	98	91	91	91
Fischerova et al. [79] *	2022	Prospective	67	86	88	93	78	87
CT								
Tempany et al. [104] *	2000	Prospective	280	78	89	48	97	88
Michielsen et al. [109] *	2014	Prospective	32	61	86	72	78	76
Schmidt et al. [54] *	2015	Prospective	15	90	91	91	90	90
Alcazar et al. [102] *	2019	Prospective	93	60	94	76	88	86
Rizzo et al. [53]	2020	Prospective	92	58	88	78	75	76
Fischerova et al. [79] *	2022	Prospective	67	80	58	80	88	82
WB-DWI/MRI								
Michielsen et al. [109] *	2014	Prospective	32	89	92	88	93	91
Garcia Prado et al. [111]	2019	Prospective	50	84	89	72	92	89
Rizzo et al. [53]	2020	Prospective	92	76	87	80	83	82
Fischerova et al. [79] *	2022	Prospective	67	89	79	89	86	88
PET/CT								
Kitajima et al. [112]	2008	Prospective	40	69	97	80	96	94
Hynninen et al. [113]	2013	Prospective	41	57	89	91	50	64
Michielsen et al. [109] *	2014	Prospective	32	48	89	73	74	73
Schmidt et al. [54]*	2015	Prospective	15	93	96	96	94	95
Feng et al. [114]	2021	Prospective	43	73	85	84	75	79

CT, computed tomography; PET/CT, positron emission tomography combined with computed tomography; PPV, positive predictive value; NPV, negative predictive value; WB-DWI/MRI, whole-body diffusion-weighted magnetic resonance imaging. * Studies that used the same cohort of patients to evaluate the different imaging methods.

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
