# Peer review of "Prediction of Surgical Outcome in Advanced Ovarian Cancer by Imaging and Laparoscopy: A Narrative Review"

_cancers, 2023, doi:10.3390/cancers15061904_

Round 1
Reviewer 1 Report
I must congratulate the authors for producing a very useful and comprehensive review of the imaging modalities and their usefulness in the assessment of patients with advanced ovarian cancer (AOC). However, i wonder if the authors should consider including in their narrative the comments i raise below to recalibrate what we are trying to achieve with pre-operative imaging.
Speaking as a senior sub-specialty gynaecological oncologist, the questions i would ask myself in the presence of a patient with AOC are:
1) Does "non-resectability" equate with "unnecessary" laparotomy? If "non-resectability" is defined as the inability to reach a zero residual (R0) and "unnecessary" laparotomy is defined as debulking effort that does not confer any survival benefit to the patient, then the authors need to persuade the reader that there is convincing evidence (in modern surgical practice) that neoadjuvant chemotherapy alone is equivalent to sub-optimal surgery and chemotherapy (in neoadjuvant or adjuvant setting). In other words, the addition of surgery for this patient has been a waste of time and has not improved their survival. If such evidence does not exist convincingly then is it ethical to withhold the surgical modality from any patient by searching for patients by pre-operative imaging who would be "unresectable". The goal of the clinician or the MDT should be to search for patients that are not going to benefit from surgery in terms of prolongation of survival. It would be incorrect to extrapolate that an unresectable patient is also the same patient who will not have any survival benefit from surgery.
If the authors are aware of any convincing evidence that equates "unresectable" with "unnecessary" laparotomy then this needs to be included in the narrative to serve as a justification for this strategy of pre-operative imaging.
It seems that the gynae-onc surgical community has become obsessed with wanting to achieve a high R0 rate by wanting to select a cohort of patients where this is possible by pre-operative imaging when in actual fact the goal should be to discriminate the cohort of patients that will benefit versus those who will not benefit from surgery in terms of survival. The arguments for accurate pre-operative imaging need urgent recalibration.
The role of pre-operative imaging should be to try and reduce morbidity of treatment if the question about survival benefit cannot be adequately answered. To this end, the role of imaging is partly to decide who will benefit from Primary debulking surgery (PDS) and those who would be better served by neoadjuvant chemotherapy to reduce the tumour burden followed by interval debulking surgery (IDS).
2) Does "non-resectability" carry the same importance in PDS compared with IDS? I suspect that many gynae-onc surgeons would argue that the goal of R0 is more important with IDS than with PDS. Does pre-operative imaging have differing accuracy in the 2 settings? A commentary on this may be useful. Decisions are often arbitrarily taken during MDT meetings that a patient after 3 cycles of neoadjuvant chemotherapy has not responded adequately on the interval imaging and that any debulking effort should be postponed to after 6 cycles or not at all. Is there any evidence in the literature that such decisions are justified based on imaging? A commentary on this may also be useful.
Reviewer 2 Report
The authors reviewed imaging methods and diagnostic laparoscopy for prediction of surgical outcome in advanced ovarian cancer. They discussed the ability of imaging and laparoscopy to predict non-resectable disease in patients with advanced ovarian cancer. They concluded that final decision for type of treatment is a multidisciplinary decision. And more accurate prediction of residual disease may be available in the feature.
The most important point of this article is introduction and usage of diagnostic imaging. This article is informative for gynecologists and radiologists and other researchers.
It might be improved on the following issues.
1. 2.3.1 Ultrasound P.12, line 307 and 2.3.2 Computed tomography: P.15, line 396: “Ultrasound-guided Tru-cut biopsy” “may be carried out under CT guidance”: Dose histologic diagnosis change the used equipment or the reading of the images?
2. Conclusions, P.20, line609: “meanwhile a careful imaging assessment of sites critical for ovarian cancer surgery remains the most useful approach”: In practice of Gynecology, is there a current flowchart for diagnostic imaging process? In what order should these modalities be used?
